

# Leveraging deep learning and ensemble learning for air quality forecasting in smart urban environment

Hafiz Muhammad Qadir[1,2], Muhammad Taseer Suleman[3], Rafaqat Alam Khan[2], Jianqiang Li[1,4], Tariq Mahmood[5] and Tanzila Saba[5]

[1] Faculty of Information Technology, Beijing University of Technology, Beijing, China
[2] Department of Software Engineering, Lahore Garrison University, Lahore, Pakistan
[3] Department of Computer Science, Bahria University, Lahore, Pakistan
[4] Beijing Engineering Research Center for IoT Software and Systems, Beijing, China
[5] Artificial Intelligence and Data Analytics (AIDA) Lab, CCIS, Prince Sultan University, Riyadh, Saudi Arabia

## ABSTRACT

Urban pollution has become a significant issue for the whole world, specifically for underdeveloped nations. This pollution poses significant challenges to public health, economic stability and environmental sustainability. The rapid growth of urbanization and industries, and inadequate regulatory frameworks has led to the deterioration of air, contamination of water and soil pollution. Major urban centers such as Lahore remain at the top among the most polluted cities, globally, with adverse effects such as rising respiratory diseases, contaminated water supplies and environmental degradation. The countries have proposed various policies and regulatory framework; however, these attempts do not reverse the trend of exacerbating urban pollution due to the lack of monitoring and measurable goals. This research proposes deep learning and ensemble learning approach to track pollution levels efficiently that could be utilized for policymaking and governance, supporting real time monitoring and data driven interventions. The findings indicate decision tree and random forest gave the most reliable and accurate air quality prediction, achieving an accuracy of 0.99 and 0.98, respectively, for particulate matter 2.5 (PM2.5) and particulate matter 10 (PM10), with high precision in classification across all categories. The smog-predict app has been made available *via* a user-friendly webserver at: https://smog-pred.streamlit.app.

# INTRODUCTION

Air pollution is one of the most serious concerns for the whole world. This growing pollution problem has emerged as a global concern, posing serious risks to both the environment and public health. The issue is getting worse and more intensified in the developing nations where governance struggles to keep up with the growth of cities. According to World Health Organization (WHO), 4.2 million deaths per year accounted for air pollution particularly in the underdeveloped countries (*World Health Organization, 2020*). Urban air pollution is fueled with various factors, such as rapid urbanization and industrialization have increased the pollution levels significantly. Pakistan faces the severe

Corresponding author
Muhammad Taseer Suleman, taseer.suleman11@gmail.com

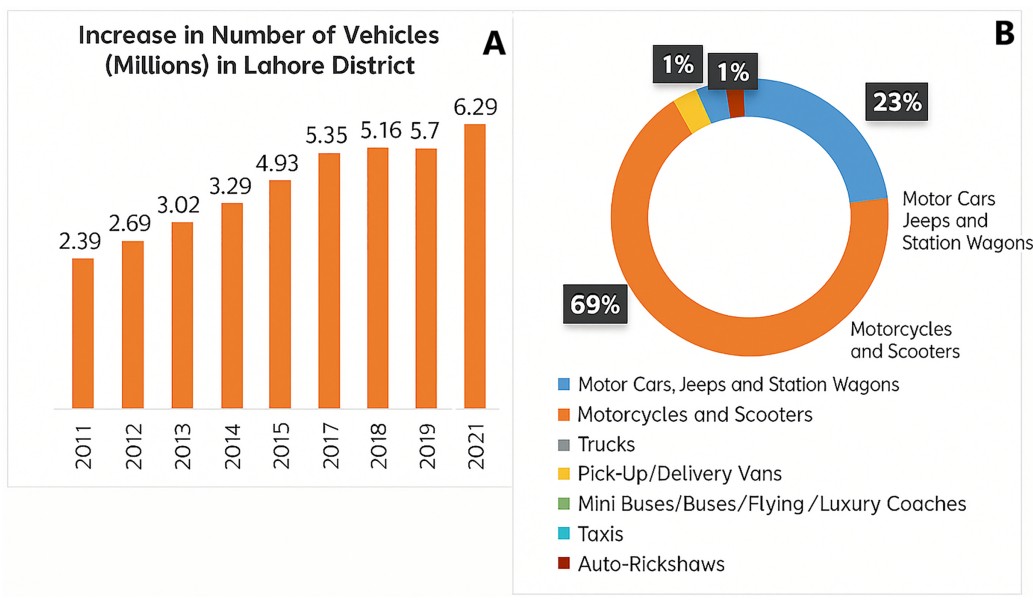

**Figure 1 Vehicles statistics in Urban Lahore (A). Increase in registered vehicles (2011–2021) in Lahore (B).** Proportion of vehicle categories in Lahore (*The Urban Unit, 2023*).

challenges of urban air pollution in cities like Lahore, Karachi, and Peshawar with Lahore consistently stands second among the top five most polluted cities in the world. The transport sector in Lahore is responsible for approximately 83% of the total emissions. Between 2011 and 2021, the number of registered vehicles in Lahore saw a drastic increase, especially two-stroke vehicles as shown in Fig. 1. Two-stroke vehicles contributed an estimated 104.76 Gg of emissions, followed by motorcars, jeeps, and station wagons at 16.34 Gg (*The Urban Unit, 2023*). The fuel quality used in Pakistan's vehicles falls under the Euro-II standard, which is considerably lower than Euro-VI standards, causing higher levels of pollutants such as nitrogen oxides and carbon monoxide. Traffic congestion further compounds emissions, as vehicles emit 3.6 times more NOx and 25 times more carbon per kilometer than the average vehicle in the United States of America (USA). Waste management practices in Lahore are a significant source of pollution, with around 3.6% of emissions attributed to open waste burning. Lahore generates approximately 7,000 tons of waste per day, with about 30% of uncollected waste openly burned. This practice emits harmful pollutants, including methane, carbon monoxide, and particulate matter. Additionally, crop residue burning contributes around 3.9% of emissions, releasing pollutants such as carbon monoxide and sulfur oxides. The percentage of concentration of particulate matter 2.5 (PM2.5) comes from different sectors in Lahore is shown in Fig. 2 that also results in smog in winters.

PM2.5 (PM ≤ 2.5 microns) and particulate matter 10 (PM10) (≤10 microns) are major air pollutants in urban areas, primarily originating from vehicle emissions, industrial activities, and construction dust. PM2.5 is particularly harmful as its tiny particles penetrate deep into the lungs and bloodstream, causing respiratory and cardiovascular

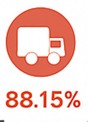 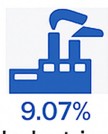 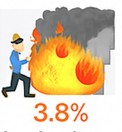 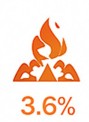 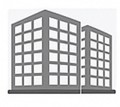 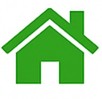

**88.15%** Transport  **9.07%** Industries  **3.8%** Agriculture (Crop Residue Burning)  **3.6%** Waste Burning  **0.14%** Commercial  **0.11%** Domestic

**Figure 2 Percentage of concentration of PM2.5 in Lahore (*The Urban Unit, 2023*).**

diseases. The WHO recommends PM2.5 levels below 5 μg/m³ (annual) and 15 μg/m³ (24-h) and PM10 below 15 μg/m³ (annual) and 45 μg/m³ (24-h) for safe air quality (*California Air Resources Board*). In Pakistan, major cities like Lahore, Karachi, and Islamabad frequently exceed safe air quality limits, with PM2.5 levels often surpassing 100 μg/m³. This leads to severe smog, health crises, and reduced visibility, exacerbated by industrial emissions, vehicular pollution, crop burning, and inadequate regulatory enforcement. The issue of air pollution requires serious intervention due to its negative impacts on human health, economy and the environment. A significant increase in diseases has been observed due to urban pollution, especially respiratory and cardiovascular diseases. It is estimated that environmental degradation costs around 5.88 % of the Gross Domestic Product (GDP) of Pakistan (*World Bank, 2024*). Pakistan has passed various legislations and implemented policies like National Clean Air Policy, 2023, Policy on Controlling Smog, 2017, National Environmental Policy, 2005 to reduce urban air pollution. However, these attempts do not reverse the trend of exacerbating urban pollution, including the quality of life and health among habitats. The main reason is the enforcement mechanism that requires monitoring and measurable goals. The country needs to invest in modern technologies such as internet of things and promote smart cities for real time data collection to track pollution levels efficiently. This centralized data platform could be utilized for policymaking and governance, supporting timely interventions. This study proposes a system that utilizes sensors data to monitor air quality by measuring the levels of PM2.5 and PM10. It is significant as it will help in monitoring policymaking and timely interventions.

## LITERATURE REVIEW

Urban air pollution has emerged as a critical concern across the world and many researchers have proposed solutions to take early measures to counter this growing issue. *Sharma et al. (2024)* provided a robust framework by using machine learning algorithms for the prediction of Air Quality Index (AQI) in smart cities. The study identified PM2.5, PM10, O₃, SO₂ as the dangerous pollutants affecting the cities due to high activities of industrial and vehicle emission. By combining random forest (RF) with XGBoost provided high accuracy and precision of AQI with lower error rates. Researchers (*Anitha, Malleswarao & Naidu, 2024*) found that the proposed machine learning models offers a foundational approach for baseline prediction in air quality. The multi variant linear

regression helped to find the relationship between input features and indicators, meanwhile the random forest was used to discern complex patterns and non-linear correlations within air quality data. This algorithm could also handle large datasets. The study showed that multivariate linear regression (MLR) and random forest regressor (RFR), with low error rates while high $R^2$ scores, shows robust predicting capabilities.

*Essamlali, Nhaila & El Khaili (2024)* demonstrated that particulate matter like PM2.5 and PM10 cause respiratory disorders health risks. Hybrid models with artificial neural network (ANN) demonstrated high accuracy in predicting pollutants levels in air. Support vector machine showed effectiveness in continuous data predictions, meanwhile random forest excels and showed abilities to perform both in classifications and regression.

*Dang et al. (2024)* leverages the use of artificial intelligence (AI) and Internet of Things (IoT) to construct the models and analyze the large datasets to understand the relationship between economic development and air pollution. This study created the theoretical model using data from urban areas for collecting weather patterns; pollutants in air *etc.*, to enable IoT sensors for intelligent health system along economic growth. Support vector machine (SVM) algorithm known for its precise handling both in classification and regression effectively handles the complex process. The study uses multidisciplinary approach by integrating the advanced data, economics analytics and environmental science to examine the complexities by offering practical framework. IoT sensors are placed in urban panels for the purpose of collecting data of weather patterns and pollution levels. The sensors allow for real-time monitoring. *Molina-Gómez, Díaz-Arévalo & López-Jiménez (2021)* showed the pivotal role of machine learning in analyzing and forecasting air quality. The simulations showed ANN, SVM and decision tree revealed that models make it simple to monitor the behavior of air pollution and offer early warnings for sustainable environment and handle the complex correlation between environmental, social and economic indicators. The study also highlighted the gaps for the purpose of sustainability assessment like unavailability or limited data and the new approach or methodology needed for accurate predictions. *Méndez, Merayo & Núñez (2023)* emphasized the dire importance of machine learning and deep learning strategies in detection of pollutants concentrations and finding air quality trends. The analysis highlighted the key findings such as long short-term memory (LSTM), multi-layer perceptron (MLP) and convolutional neural network (CNN) algorithm for time-series forecasting and particularly for pollutants such as PM2.5 and AQI. The study also found that by combining pollutants data and meteorological variables such as wind speed, temperature and humidity enhanced the efficiency and accuracy in predictions.

*Liao et al. (2020)* put the emphasis by overcoming the limitations of traditional model like chemistry transport model and statistical techniques, the deep learning has a great deal and significant advantages in improving the air quality forecasting. In extracting complex, high dimensional and nonlinear features the DL architecture such as CNN, recurrent neural network (RNN) and LSTM have shown its success. These models showed the proven capacity in managing large datasets and filling the gaps efficiently. Furthermore, the deep learning (DL) models outperform the traditional techniques and perform better than conventional models in integrating satellite and ground-level data. *Kaur et al. (2023)*

conducted a systematic review emphasizing the significance of DL models in identifying both spatial and temporal correlation dependencies in data for predicting air quality. It finds out hybrid models (CNN, RNN, LSTM) are the most effective technique in handling air pollution datasets. In this systematic review, a variety of evaluation metrics applied to analyze the performance of DL models for different set of metrics used depending on various models like for time series and regression model, error based metrics used.

*Gugnani & Singh (2022)* indicated that LSTM based models had been performing better than temporal forecasting and were particularly good at sequential data. Spatiotemporal hybrid models based on CNN and LSTM, while not very old, demonstrate great results in terms of accuracy, and while they require more development, sophisticated frameworks using attention mechanisms and graph convolutional networks, or GCN, demonstrate great potential. While they suffer from vanishing gradient problems, recurrent neural networks (RNN) are suitable for sequential data.

*Zaini et al. (2022)* emphasized that deep learning outperforms traditional statistical and machine learning methods. Specifically, LSTM and gated recurrent unit (GRU) should be able to solve time series dependent problem proficiently. CNN are rather efficient at feature extraction as well as spatial data processing. Benefits of hybrid models such as deep learning models with auxiliary methods solving problems by improving accuracy. CNN-LSTM is one of the spatiotemporal forecasting methods while optimization algorithms are used to update parameters.

*Ansari & Quaff (2025a)* analysis of data on air quality in the Azamgarh district shows that temporal factors are greatly influential and especially during the winter season, which also has bad weather and serious pollution caused by burning of biomass. With the dataset of 8,760 hourly observation data and considering the hourly simulation of the timed Air Quality Index (AQI), six deep learning models such as feedforward neural network (FNN), CNN, LSTM, GRU, multilayer perceptron (MLP), and Transformer were employed to evaluate the performance in the series of timed data pattern during a year period (July 2022–June 2023). Further, the analysis of the variance through MANOVA, ANOVA and t-tests determining trends of pollutant concentrations detected higher AQI at night and during winter. From the models the FNN showed the highest level of fitness with an mean absolute error (MAE) of 2.89, root mean square error (RMSE) of 4.99 and R-square of 0.9971.

*Kang et al. (2018)* highlights the importance of the development of real-time air quality assessment systems for smart cities, underlining big data and complex machine learning methods as the basis for accurate spatial-temporal predictions. Through such techniques mentioned, for example artificial neural networks, decision trees, as well as the genetic algorithms, it demonstrates that they offer a degree of sophistication to emulate complex pollution behaviors, and estimate IoT networks data, satellite imagery, and even sensor-based systems data capabilities. Despite these contributions, basic research questions continue to limit the credibility of data, the reliability of sensors, and the extensibility of models. The application of modern algorithms including GA-ANN and random forests has proven to improve the efficacy of pollutant predication, as well as improve the urban AQI forecast as compared to conventional techniques. The outcomes

suggest considering the future research of multi-modal and time-variable system approaches to fill the existing gaps and improve the air quality forecast applying to emerging smart city environments further.

*Joharestani et al. (2019)* provides confirmation of the applicability of machine learning in air quality prediction, most especially when diverse and quality input features are used. Empirical with Tehran, 37 stations including Satellite imagery Aerosol Optical Depth (AOD), Met data, Geographical information gathered to predict PM2.5 levels for 4 years. Data normalization through attribute selection by employing rational approaches followed by model tuning yielded a superior model of Extreme Gradient Boost (XGBoost) with an R-squared estimate of 0.81, MAE of 10.0 $\mu g/m^3$, and RMSE of 13.62 $\mu g/m^3$. Random forest and deep learning models also showed good stability in their predictions and their R-squared respectively were 0.78 and 0.77.

*Huang & Kuo (2018)* presents APNet, a deep learning model employing CNN for feature learning and LSTM networks for temporal dynamics to enhance the rapidly growing PM2.5 forecasting precision. In the experiments with Beijing's PM2.5 data, APNet achieved better results than benchmark models, including support vector machine, random forest, and decision trees, when it was trained using features including the PM2.5 concentration, wind speed, and rainfall. In other words, prediction accuracy was at its best with the lowest RMSE, MAE, and MAPE values while the highest Pearson correlation coefficients and IA tests for the employed model.

*Cheng et al. (2018)* presents ADAIN: Attentional Deep Air Quality Inference Network, a new model for estimating the quality of the air in large cities, especially where there are no fixed monitoring stations. ADAIN utilizes both feed-forward and recurrent neural networks and applies an attention mechanism to flexibly control the POI information, road network, meteorological condition data, and historical air quality data from multiple stations. This approach improves the model's forecast functionality and goes beyond simple proximity-based techniques.

*Zhu et al. (2018)* showed the study of temporal dependencies and the efficient deployment of parameters in an ML model for air pollution projection is underlined. Specifically, as $O_3$, PM2.5, and $SO_2$ hourly monitoring data from Chicago were adopted for analysis, the authors considered a multi-task learning (MTL) approach since it was hypothesized that tasks constructed for optimization will provide better accuracy while the complexity of the model is relatively small. The core meteorological variables and U.S. EPA data were pre-processed to handle the missing values within the variables and rescaled where necessary. Pre-processing methods such as Frobenius norm, nuclear norm and consecutive close were integrated to improve model structure. The analysis of the results revealed that the proposed light MTL models can be from 8% to 15% more accurate in terms of RMSE, compared to the baseline models, and more accurate for those pollutants with similar day/night curves.

*Pande et al. (2025)* evaluated the prediction of Air Quality Index (AQI) in Delhi using three machine learning models: linear regression, decision tree (DT), and random forest (RF). Historical air pollution data (PM2.5, $NO_2$, $SO_2$, and $O_3$) from 1987 to 2020 was cleaned and analyzed across three scenarios with six input variables. The model stability

along with accuracy was achieved through 10-fold cross-validation while $R^2$ and RMSE evaluated the performance output. RF surpassed DT and linear regression by achieving highest $R^2$ along with minimum RMSE values in the experiments. Data classification with DT was efficient but RF achieved superior results than DT and linear regression yielded increased error rates when giving input predictions. The model accuracy increased with the addition of essential pollutants including PM2.5 and $NO_2$. These findings underscore RF's adaptability to large datasets and highlight the potential of machine learning in air quality management, offering policymakers robust tools for developing pollution control strategies.

_Tong et al. (2019)_ established deep learning technology can interpret air pollution through bidirectional long short-term memory (bi-LSTM), recurrent neural networks (RNNs) which produces better PM2.5 distribution understanding. The model application used time and space information together to achieve better PM2.5 interpolation results than basic unidirectional LSTM models. Researchers tested model performance by applying PM2.5 Florida data from 2009 which came from the U.S. EPA and measured their results using MAE, RMSE, MAPE alongside parameter optimization through cross-validation. The research established that time-based factors surpass spatial relationships due to their essential impact on air pollution measurement.

_Shahsavani et al. (2025)_ investigated the concentrations of heavy metals of PM10 aerosols near Bakhtegan Lake and health risks associated with them. The concentration of 22 metals in air from a neighboring village was measured by inductively coupled plasma mass spectrometry, while random forest machine learning algorithms were used to estimate nickel concentrations. Sources of nickel pollution were found to include copper and lead. PM10 concentration in the study was $78.12 \pm 24.56$ µg/m$^3$ and similarly showed breaching the WHO recommended standards at 24-h mark which is 50 µg/m$^3$. Concentration of arsenic, manganese, and nickel exceeded WHO permissible limits was due to natural contributions from crust and sediment and anthropogenic contributions from industrial emissions, motor vehicles, and combustion.

_Tawiah, Daniyal & Qureshi (2017)_ highlights the demand for new and sectorial strategies in decreasing $CO_2$ by comparing statistical models like ARIMA and exponential smoothing with the MLP neural networks with an application of $CO_2$ emissions of Pakistan in energy, manufacturing and transport sectors for the time series 1971–2014. Different performance measurements such as MAPE and sMAPE were used to measure accuracy based on the data used, projections were made up to the year 2030 for policy making. According to the findings, the trend in $CO_2$ emissions is on the increase with forces from the energy sector taking the largest cut from the industrial and transport sectors. Neural network models gave better results as compared with statistical techniques involved were more accurate in terms of approximation.

_Ameer et al. (2019)_ evaluated air quality prediction in smart cities using four machine learning techniques, DT, RF, gradient boosting, and MLP. The monitored PM2.5 concentrations and related meteorological parameters from five Chinese cities for 5 years (2010–2015) were evaluated using modeling accuracy indices such as RMSE and MAE. Among the models, we discovered that RF regression offered the highest accuracy and the

least amount of time spent to make predictions while hauling least error rates. Although DT regression had a less complex and shorter time to solve, it provided larger error bounds.

*Ansari & Quaff (2025b)* evaluated the hourly AQI for the city of Azamgarh, India for July 2022 to June 2023 using air pollutant concentrations and meteorological conditions gathered by a sensor network of Pollutrem's PM2.5, PM10, $NO_2$, and $SO_2$ concentrations and temperature, humidity, wind speed, and UV radiation. Ten-fold cross validation was used to train eight machine learning models namely XGBoost and CatBoost for the estimation of AQI, which resulted in the estimate of AQI 123 which is interpreted as moderately polluted air. The best model was identified as XGBoost; the model yielded an RMSE of 0.32 and took a computational time of 1.61 s. The results of sensitivity analysis further indicated that PM2.5, PM10, $NO_2$ and $SO_2$ had the highest impact on AQI.

*Abdulraheem et al. (2025)* showed PM2.5 concentrations and their trends across the eleven selected cities in Nigeria were explained by precipitation, temperature, nighttime lights and population density from 2000–2020. They applied linear regression, K-nearest neighbors, decision tree regression, support vector regression, neural network, CatBoost algorithm; Five-fold cross validation for the assessment and statistical assessment such as $R^2$, RMSE, MAE, MAPE are also used. CatBoost was the winner of the proposed models, followed by an effective way to work with categorical data and no sign of over-fitting during its training phase, and, on the opposite side, decision tree regressor exhibited the worst performance. Table 1 shows the comparative analysis of related work with current research work.

## RESEARCH METHODOLOGY

The proposed methodology employs deep learning and ensemble learning algorithms to predict the levels of PM2.5 and PM10. Following are the breakdowns of the detailed methodology starting from data preprocessing to the simulation of the results.

### Dataset description

The dataset consists of 26,746 total samples collected between 2020 and 2023, with an 80:20 train-test split. Each year has a substantial number of records, with 2021, 2022, and 2023 contributing significantly (~8,600+ records each), while 2020 has a smaller dataset (887 samples). The train-test division ensures that 21,397 samples are used for training, while 5,349 are allocated for testing, maintaining a robust dataset for model evaluation. Table 2 shows the detailed description of the dataset and its divisions. The dataset used in our study was obtained through a weather API and includes various atmospheric pollutants such as CO, NO, $NO_2$, $O_3$, $SO_2$, PM2.5, PM10, and $NH_3$, along with the AQI and a derived PM10 smog level. While direct meteorological parameters like temperature, humidity, and wind speed were not available in this dataset, the pollutant concentration levels implicitly reflect environmental conditions, as these are often co-dependent on weather phenomena.

**Table 1 Comparative analysis of related work and current study.**

| Study/Author | Key focus/Contribution | Techniques used | Limitation/Scope | Difference with current study |
|---|---|---|---|---|
| *Sharma et al. (2024)* | AQI prediction in smart cities | RF + XGBoost | Lacks deep models | Our study compares DL & ML across 4 years |
| *Essamlali, Nhaila & El Khaili (2024)* | Health risk *via* PM2.5/PM10 | ANN, SVM, RF | Focus on health impacts | Our work focuses on temporal performance |
| *Dang et al. (2024)* | AI + IoT for eco-health | SVM, IoT sensors | Theoretical framework | Our study uses real pollutant datasets |
| *Molina-Gómez, Díaz-Arévalo & López-Jiménez (2021)* | ML for sustainable environment | ANN, DT, SVM | Highlights of early warning gaps | We assess long-term yearly performance |
| *Méndez, Merayo & Núñez (2023)* | DL + meteorological features | LSTM, CNN, MLP | High complexity models | Our study compares DL *vs* ML without external features |
| *Kaur et al. (2023)* | DL in spatial-temporal data | CNN, LSTM, RNN | Systematic review | We provide empirical validation with Lahore data |
| *Ansari & Quaff (2025a)* | Hourly AQI with 8,760 samples | FNN, GRU, LSTM | Focus on fine-grained hourly data | We focus on yearly accuracy patterns |
| *Huang & Kuo (2018)* | CNN + LSTM (APNet) | Hybrid DL | Beijing PM2.5 only | We include PM2.5 & PM10 across multiple years |
| *Cheng et al. (2018)* | ADAIN: attention-based DL | RNN, FFN | Limited to cities without stations | Our dataset includes fixed sensor locations |
| *Abdulraheem et al. (2025)* | 20-year PM2.5 trend in Nigeria | CatBoost, SVR | Focus on spatial trends | Our focus is temporal model comparison in Lahore |

**Table 2 Smog dataset description with training and test samples.**

| Year | Total samples | Train samples (80%) | Test samples (20%) |
|---|---|---|---|
| 2020 | 887 | 709 | 177 |
| 2021 | 8,693 | 6,954 | 1,738 |
| 2022 | 8,563 | 6,850 | 1,712 |
| 2023 | 8,603 | 6,882 | 1,720 |
| 2020–2023 | 26,746 | 21,397 | 5,349 |

## Data preprocessing

The air quality dataset was sourced from an Excel file containing PM2.5 and PM10 AQI records. Missing values in the 'Calculated PM2.5 AQI' and 'Calculated PM10 AQI' columns were handled using linear interpolation, followed by rounding and conversion to integer values. Based on standard AQI thresholds, new columns ('PM2.5 Smog Level' and 'PM10 Smog Level') were created, assigning numerical values from 1 (Good) to 6 (Hazardous) to represent increasing pollution severity. The preprocessed dataset was then saved to a new Excel file for further analysis.

## Feature selection and label preparation

To prepare the dataset for machine learning models, independent variables (features) were separated from the dependent variable (labels), excluding the "Date" column. Labels were

remapped into a 0-based index format to maintain compatibility with machine learning and deep learning models.

## Data splitting and scaling

To facilitate robust model training and evaluation, the dataset was split into training (80%) and testing (20%) sets using stratified sampling to preserve the label distribution. A StandardScaler was applied to ensure that all input features had a mean of 0 and a standard deviation of 1, improving model convergence and performance.

## Model development and evaluation

A combination of deep learning and traditional machine learning models was implemented for smog level prediction. A CNN in deep learning models contained convolutional layers and dense layers together with max-pooling strategies to feature extraction and smog levels classification. Before training the model, we reshaped the data for CNN input while optimizing it through the combination of categorical cross-entropy loss and Adam optimizer. A deep neural network employed MLP architecture to implement multi-layer perceptron as it included dense layers and dropout regularization to boost model generalization. LSTM-based framework processed time-dependent data patterns after transforming the dataset into sequences. The research adopted a linear kernel SVM as its traditional machine learning approach for training standardized data alongside decision trees for modeling hierarchical decision rules. The implementation of random forest ensemble served to produce more accurate predictions while the K-nearest neighbors (KNN) classifier used proximity assessments of data points to generate results. Table 3 contained the details of the hyperparameters that were tuned for optimization. The assessment of prediction models occurred through primary evaluation based on accuracy measurement. Confusion matrices were generated to analyze classification performance across different smog level categories. Confusion matrices were represented as heat maps to facilitate an intuitive understanding of classification results. Smog levels were annotated according to their real-world significance (*e.g.*, Moderate, Unhealthy, Unhealthy Sensitive, Very Unhealthy and Hazardous).

## Comparative analysis and model selection

Models were compared based on accuracy and confusion matrix analysis. The optimal model was selected based on its ability to correctly predict smog levels across all categories with the highest accuracy and lowest misclassification rate. The methodology pictorially represented in Fig. 3 ensures a rigorous approach to smog level prediction by integrating classical machine learning techniques with deep learning architectures, leveraging robust preprocessing, evaluation, and interpretability techniques.

## RESULTS AND DISCUSSION

The performance evaluation of different machine learning models for PM2.5 and PM10 prediction over 4 years (2020–2023) demonstrates significant variations in accuracy across methodologies as visible in Fig. 4. Table 4 exhibits the detailed results for PM2.5, decision tree and random forest exhibited the highest predictive accuracy, achieving a perfect score

**Table 3  Tuned parameters values of models.**

| Model | Random forest | KNN | Decision tree classifier | SVM | CNN/DNN/LSTM |
|---|---|---|---|---|---|
| Hyper-parameter value (s) | *n_estimators* = 100 | *N* = 5 | *Splitter* = 'random' | *C* = 10 | *Learning rate = 0.001* |
| | *max_depth* = 50 | | *Max_depth* = 50 | *Gamma* = 0.0001 | *Batch size = 32* |
| | *max_features* = 'Auto' | | *min_samples_leaf* = 4 | *Kernel* = linear | *Epochs = 50* |
| | *min_samples_split* = 10 | | *random_state*= 'None' | *Probability* = 'True' | *Validation Split = 0.1* |
| | *min_samples_leaf* = 5 | | *min_weight_fraction_leaf* = 0.1 | *Verbose* = 'False' | |
| | | | | *Random_state* = none | |

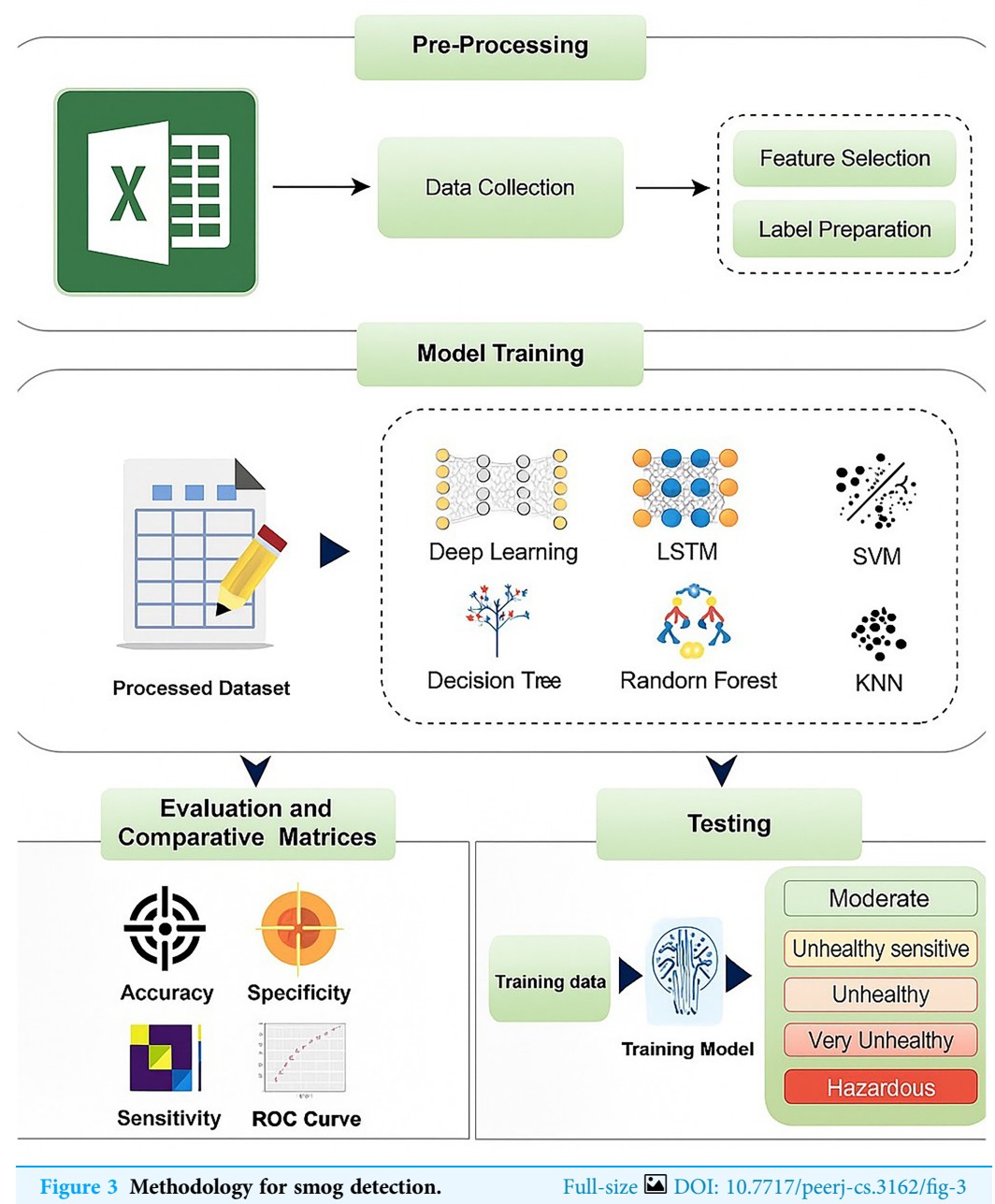

**Figure 3  Methodology for smog detection.**

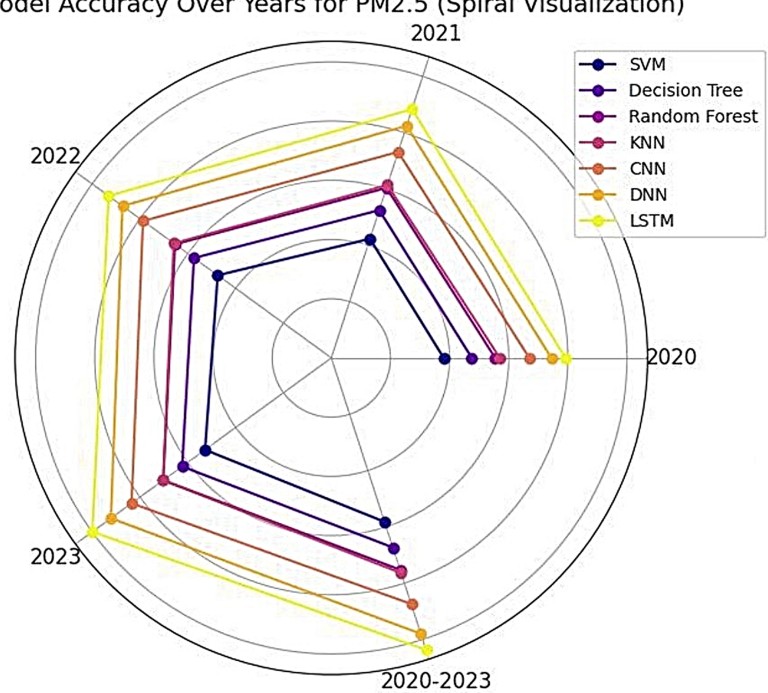

**Figure 4 Spiral visualization for PM2.5.**

**Table 4 PM2.5 results.**

| Method | Year-2020 | Year-2021 | Year-2022 | Year-2023 | Year-2020–2023 |
|---|---|---|---|---|---|
| SVM | 0.95 | 0.94 | 0.95 | 0.96 | 0.97 |
| Decision tree | 0.99 | 0.99 | 0.99 | 0.99 | 0.99 |
| Random forest | 0.99 | 0.99 | 0.99 | 0.99 | 0.99 |
| KNN | 0.89 | 0.89 | 0.89 | 0.89 | 0.91 |
| CNN | 0.93 | 0.95 | 0.96 | 0.96 | 0.95 |
| DNN | 0.93 | 0.97 | 0.97 | 0.97 | 0.98 |
| LSTM | 0.90 | 0.95 | 0.95 | 0.97 | 0.96 |

0.99 in 2021, 2022, and 2023. LSTM and SVM performed consistently well, with LSTM improving from 0.90 in 2020 to 0.97 in 2023, and SVM maintaining a stable accuracy around 0.95–0.96. CNN and DNN also showed promising results, both reaching 0.97 by 2023. However, KNN consistently recorded the lowest accuracy, stagnating at 0.89 across all years. While some models like decision tree and random forest achieved high test accuracy, the potential risk of overfitting exists. Figures 5 and 6 show the simulation of confusion matrix for concentration of PM2.5 for 2020–2023. However, to mitigate this, a validation split has been employed for deep learning models and used ensemble methods like random forest for robustness. Confusion matrices have also been analyzed to ensure consistent performance across classes. To evaluate models, F1-score, precision and recall scores were computed and presented in Table 5 for PM2.5.
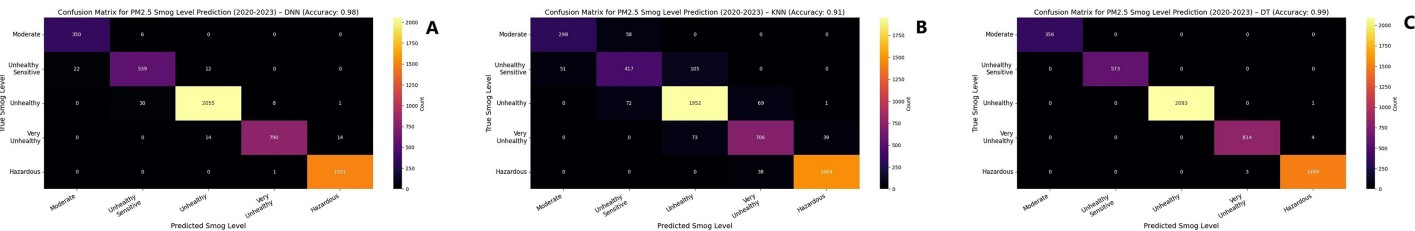

**Figure 5 Confusion matrix for concentration of PM2.5 for 2020–2023.** (A) DNN (B) KNN (C) Decision tree (DT).

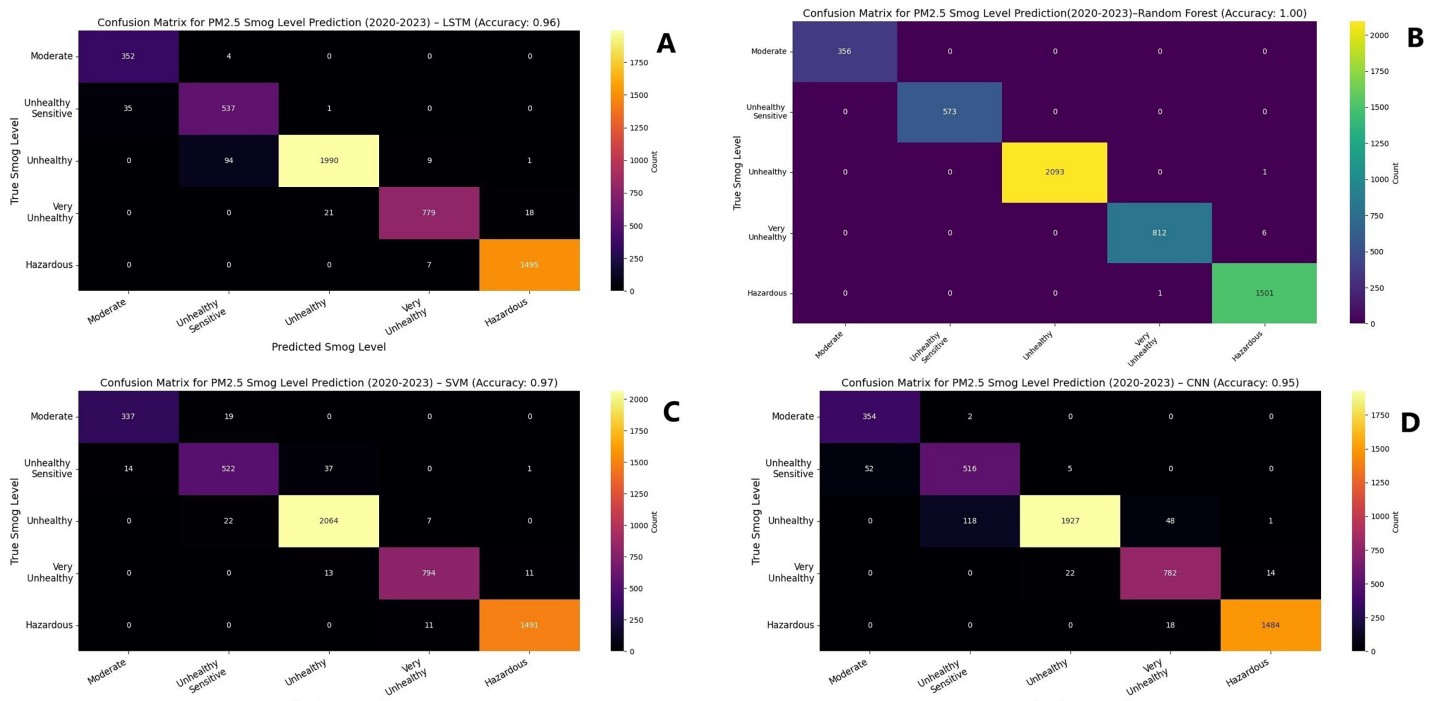

**Figure 6 Confusion matrix for concentration of PM2.5 for 2020–2023.** (A) LSTM (B) Random forest (C) SVM (D) CNN.

**Table 5 F1-score, precision and recall values for PM2.5.**

| Model | F1-score | Precision | Recall |
|---|---|---|---|
| CNN | 0.93 | 0.92 | 0.94 |
| DNN | 0.97 | 0.97 | 0.97 |
| LSTM | 0.95 | 0.95 | 0.96 |
| RF | 0.99 | 0.99 | 0.99 |
| SVM | 0.95 | 0.96 | 0.95 |
| KNN | 0.88 | 0.88 | 0.87 |
| DT | 0.99 | 0.99 | 0.99 |

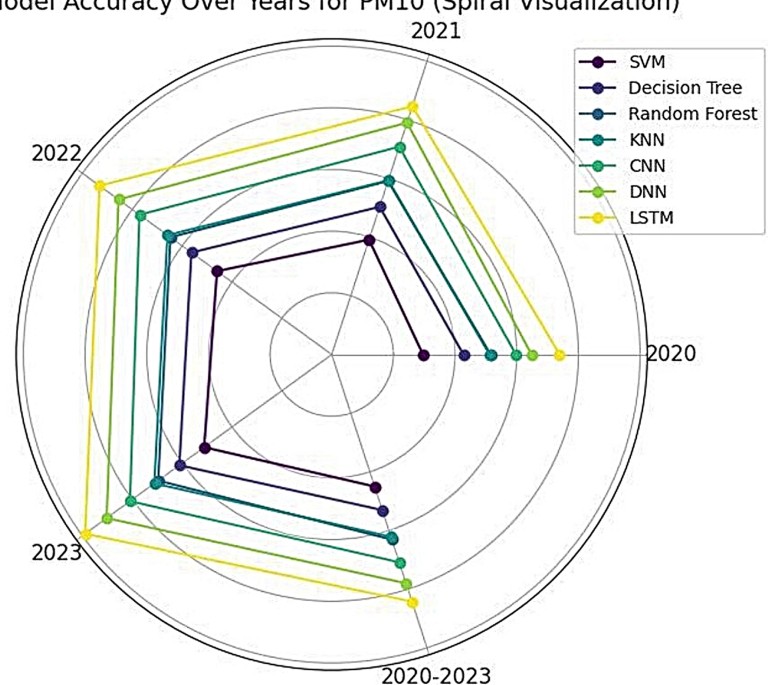

**Figure 7 Spiral visualization for PM10.**

**Table 6 PM10 results.**

| Method | Year-2020 | Year-2021 | Year-2022 | Year-2023 | Year-2020–2023 |
|---|---|---|---|---|---|
| SVM | 0.74 | 0.87 | 0.92 | 0.93 | 0.75 |
| Decision tree | 0.89 | 0.95 | 0.97 | 0.97 | 0.78 |
| Random forest | 0.92 | 0.97 | 0.98 | 0.98 | 0.83 |
| KNN | 0.80 | 0.86 | 0.89 | 0.90 | 0.74 |
| CNN | 0.83 | 0.92 | 0.94 | 0.93 | 0.77 |
| DNN | 0.81 | 0.93 | 0.95 | 0.95 | 0.78 |
| LSTM | 0.84 | 0.91 | 0.95 | 0.96 | 0.78 |

For PM10 prediction, random forest and decision tree again demonstrated superior performance, reaching an accuracy of 0.98 and 0.97, respectively, by 2023 as shown in Fig. 7. Table 6 explains the complete results LSTM exhibited a steady improvement, rising from 0.84 in 2020 to 0.96 in 2023. SVM showed a notable increase in accuracy, progressing from 0.74 in 2020 to 0.93 in 2023. CNN and DNN followed a similar upward trajectory, with both models attaining an accuracy of 0.95 by 2023. KNN, while showing improvement, remained the lowest-performing model, increasing from 0.80 in 2020 to 0.90 in 2023. To evaluate models, F1-score, precision and recall scores were computed and presented in Table 7 for PM10. These findings suggest that ensemble-based methods such as decision tree and random forest offer the highest reliability for air quality prediction, while deep learning models, particularly LSTM and DNN, exhibit strong adaptability and
**Table 7 F1-score, precision and recall values for PM10.**

| Model | F1-score | Precision | Recall |
|---|---|---|---|
| CNN | 0.75 | 0.77 | 0.75 |
| DNN | 0.76 | 0.79 | 0.75 |
| LSTM | 0.74 | 0.75 | 0.75 |
| RF | 0.80 | 0.81 | 0.80 |
| SVM | 0.73 | 0.75 | 0.73 |
| KNN | 0.75 | 0.88 | 0.65 |
| DT | 0.74 | 0.75 | 0.75 |

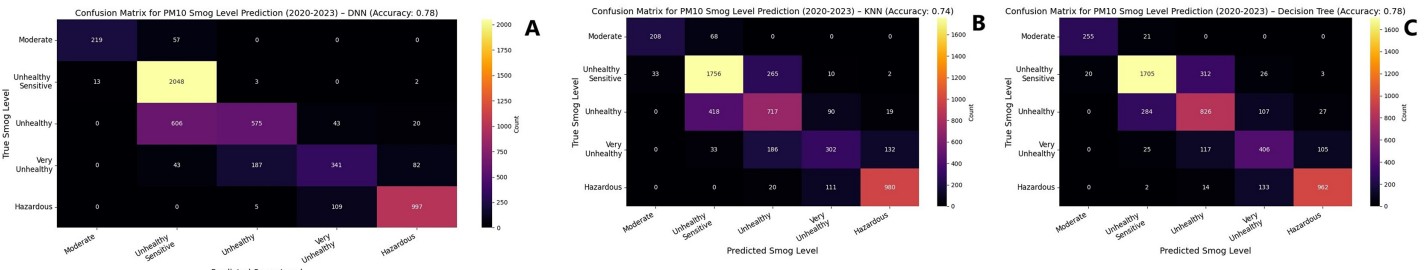

**Figure 8 Confusion matrix for concentration of PM10 for 2020–2023.** (A) DNN (B) KNN (C) Decision tree (DT).

continuous performance enhancement over time. Figures 8 and 9 represent the simulation of confusion matrix for concentration of PM10 for 2020–2023.

The study results show useful information about the quantitative assessment of different machine learning and deep learning algorithms used for estimating PM2.5 and PM10 values. The research shows ensemble models decision tree and random forest attained superior accuracy scores compared to other methods throughout all investigated years. The superior results stem from their exceptional capability to detect complex decision boundaries and deal with non-linear relationships. The predicted accuracy for PM2.5 reached 0.99 in all three tested years along with PM10 where the prediction accuracy exceeded 0.98 by 2023. These models were recommended for air quality prediction because their stable performance indicates they will deliver precise results.

Deep learning approaches, particularly LSTM and DNN, demonstrated strong adaptability over time. LSTM models showed continuous improvement of accuracy when predicting both PM2.5 and PM10 showing their strength in identifying temporal patterns in the data. Over the data of the year 2023, LSTM achieved PM2.5 accuracy at 0.97 along with PM10 accuracy at 0.96 thus establishing itself as a time-series-based air quality prediction choice. The performance of DNN increased substantially until 2023 when it achieved accuracy of 0.97 for PM2.5 and 0.95 for PM10. The increasing trend in performance for these models highlights the effectiveness of deep learning architectures in learning intricate patterns from air quality data.

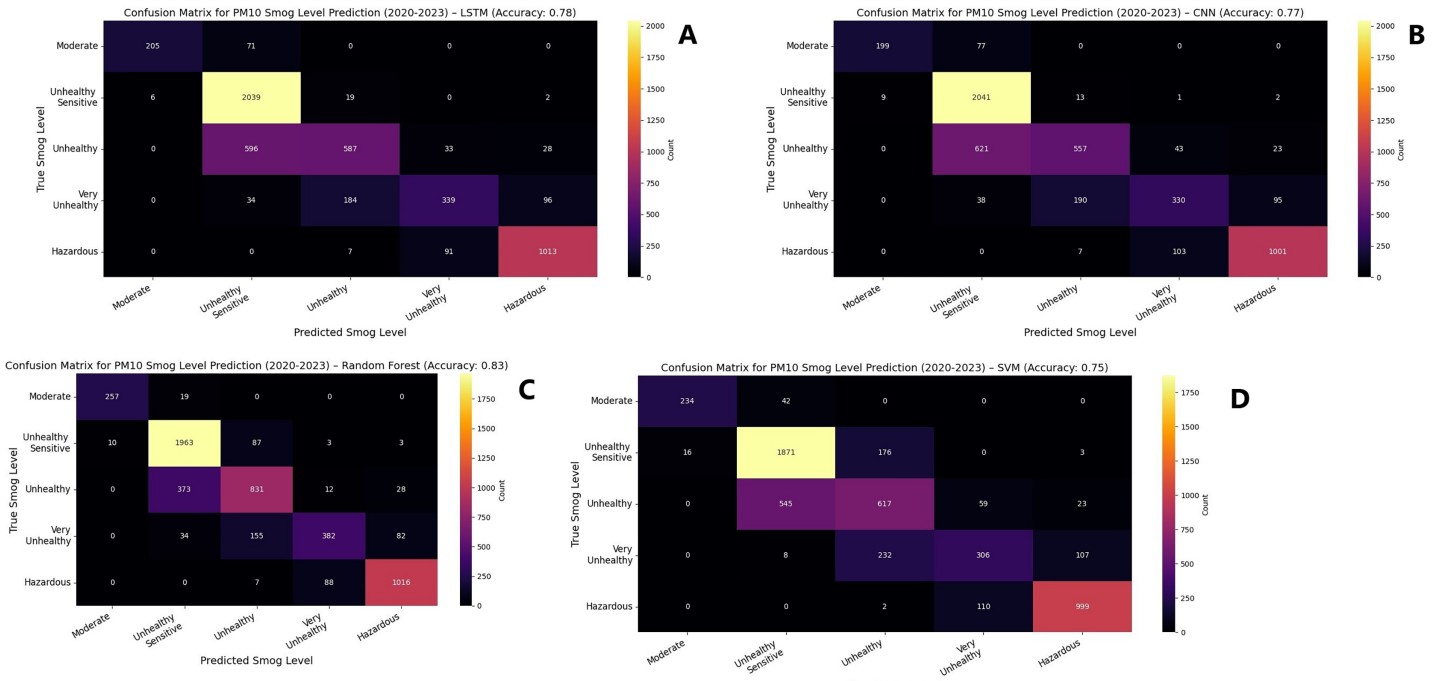

**Figure 9 Confusion matrix for concentration of PM10 for 2020–2023.** (A) LSTM (B) CNN (C) Random forest (D) SVM.

Traditional machine learning methods such as SVM and KNN exhibited varying degrees of effectiveness. The accuracy rates of SVM remained consistent for PM2.5 from 0.95 to 0.96 throughout the period alongside a substantial PM10 performance boost from 0.74 in 2020 to 0.93 in 2023. Applications of SVM demonstrate that the model performs effectively when suitable air quality features are properly selected and preprocessed. The KNN model-maintained consistency with low accuracy levels as PM2.5 reached only 0.89 throughout all years and PM10 showed slight improvement from 0.80 to 0.90 between 2020 and 2023. The substandard performance of KNN during these tasks can be attributed to its vulnerability to noise and its requirement to depend on local neighbor relationships that might not work well for complex air quality prediction challenges.

The CNN-based model forecasted PM2.5 with 0.96 accuracy as well as PM10 with accuracy of 0.93 for 2023. Years of improvement in CNN models indicate their effective capability to extract both spatial and hierarchical characteristics. The accuracy performance of CNN as a stand-alone model remained lower than ensemble methods and LSTM which implies it needs other methods to reach optimal air quality forecasting results. For PM2.5, deep learning models such as DNN and LSTM achieved high performance, with DNN reaching accuracy of 0.98, over the full 2020–2023 period, comparable to traditional models like decision tree and random forest (both 0.99). In the case of PM10, random forest delivered the best overall accuracy (0.83 across all years), but deep models like LSTM (up to 0.96), DNN (0.95), and CNN (0.94) also performed competitively in individual years. These results suggest that while deep learning does not

**Table 8 Model accuracy scores with 95% confidence intervals.**

| Model | Accuracy | 95% Confidence interval |
|---|---|---|
| Random forest | 0.83 | [0.82–0.84] |
| DNN | 0.78 | [0.77–0.79] |
| LSTM | 0.78 | [0.77–0.79] |
| Decision tree | 0.78 | [0.77–0.79] |
| CNN | 0.77 | [0.76–0.78] |
| SVM | 0.75 | [0.74–0.76] |
| KNN | 0.74 | [0.73–0.75] |

**Table 9 Paired t-test p-values for model comparison.**

| Comparison | p-value | Significance ($p < 0.05$) |
|---|---|---|
| CNN *vs* DNN | 0.0002 | ✓ |
| CNN *vs* LSTM | 0.0002 | ✓ |
| CNN *vs* SVM | 0.0000 | ✓ |
| CNN *vs* Decision tree | 0.4585 | ✗ |
| CNN *vs* Random forest | 0.0000 | ✓ |
| CNN *vs* KNN | 0.0000 | ✓ |
| DNN *vs* LSTM | 0.8273 | ✗ |
| DNN *vs* SVM | 0.0000 | ✓ |
| DNN *vs* Decision tree | 0.4518 | ✗ |
| DNN *vs* Random forest | 0.0000 | ✓ |
| DNN *vs* KNN | 0.0000 | ✓ |
| LSTM *vs* SVM | 0.0000 | ✓ |
| LSTM *vs* Decision tree | 0.3968 | ✗ |
| LSTM *vs* Random Forest | 0.0000 | ✓ |
| LSTM *vs* KNN | 0.0000 | ✓ |
| SVM *vs* Decision tree | 0.0004 | ✓ |
| SVM *vs* Random forest | 0.0000 | ✓ |
| SVM *vs* KNN | 0.0567 | ✗ (borderline) |
| Decision tree *vs* RF | 0.0000 | ✓ |
| Decision tree *vs* KNN | 0.0000 | ✓ |
| Random forest *vs* KNN | 0.0000 | ✓ |

always outperform simpler models, it remains effective, particularly when modeling year-specific patterns in air quality data. To validate model comparison, 95% confidence intervals were computed for accuracy and performed paired t-tests. Random forest consistently outperformed other models ($p < 0.0001$), while differences between DNN and LSTM were not statistically significant ($p = 0.8273$). Confidence intervals also supported these findings, with a minimal overlap between higher- and lower-performing models. Tables 8 and 9 contained the values for 95% confidence intervals and paired t-test p values respectively.

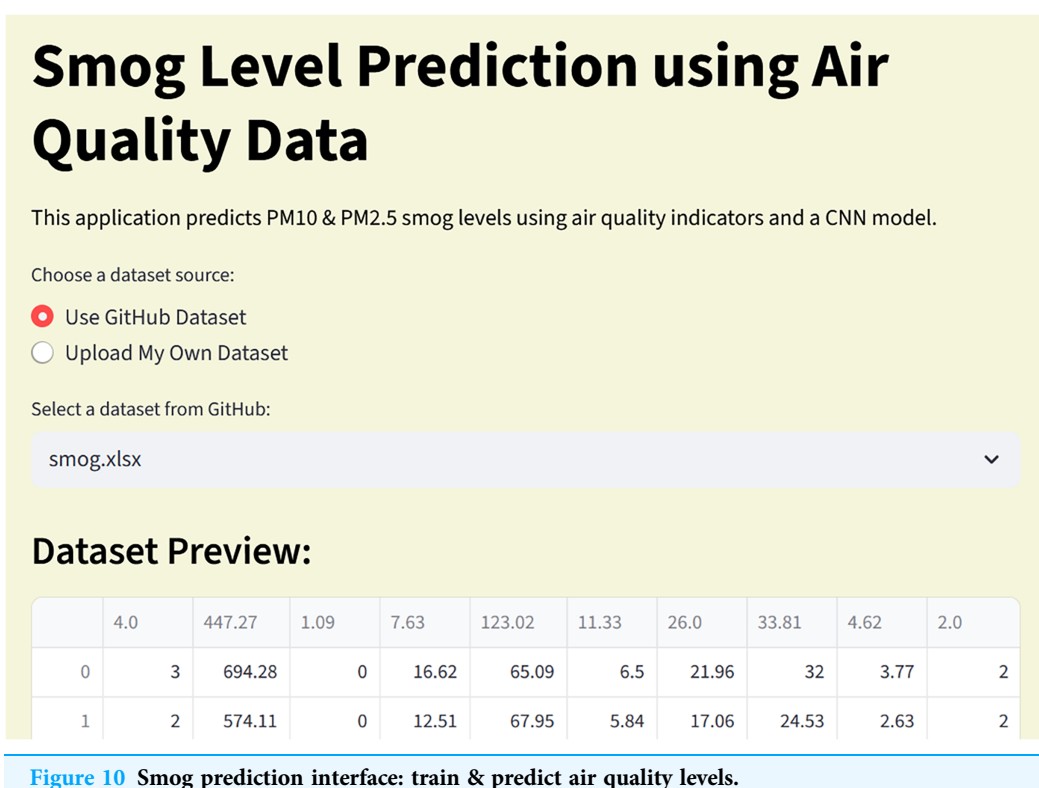

**Figure 10 Smog prediction interface: train & predict air quality levels.**

Overall, the study highlights the effectiveness of ensemble-based techniques such as decision tree and random forest for high-accuracy air quality prediction. Meanwhile, deep learning models, especially LSTM and DNN, exhibit strong potential for long-term predictive performance. The findings reinforce the importance of selecting appropriate models based on the specific requirements of air quality forecasting, such as real-time predictions, adaptability to temporal patterns, and classification accuracy. Future work may explore hybrid approaches that combine ensemble learning with deep learning to further enhance predictive capabilities and provide more accurate and reliable smog level forecasts. The data is region-specific and lacks certain meteorological features (*e.g.*, temperature, humidity), which may limit model generalizability. The incorporation of multi-regional data and weather attributes in future studies to improve robustness.

## WEBSERVER DEVELOPMENT

A user-friendly webserver has been developed for the research community, to validate the performance of proposed model. Figure 10 shows the interface of app which has made available public *via* a user-friendly webserver at: https://smog-pred.streamlit.app.

## CONCLUSION

This study aims to demonstrate a real time air quality monitoring system that detects levels of key pollutants like PM2.5 and PM10. This could be helpful for the decision makers to develop measure able goals and action plan to counter the hazardous impact of urban air

pollution. This study utilized deep learning and ensemble-based models *i.e.*, deep learning, LSTM, decision tree, random forest, SVM and KNN. The experiments were conducted on the data of consecutive 4 years which showed the effectiveness of decision tree and random forest as the most reliable and accurate for air quality prediction, achieving an accuracy of 0.99 and 0.98, respectively, for PM2.5 and PM10, with high precision in classification across all categories. In future, the dataset will be enhanced, and more techniques will be explored and employed to get more reliable results for the authentic and efficient early warnings to counter the disastrous impact of urban air pollution. The model can be enhanced by integrating temperature, humidity, and wind-related features to improve generalization and accuracy under varying weather patterns.

## ACKNOWLEDGEMENTS

Prince Sultan University, Riyadh, Saudi Arabia, provided access to laboratory facilities, computing resources, and technical assistance during the research and manuscript preparation stages.

### Funding
This study is supported by the National Key R&D Program of China with project no. 2023YFB2704601. The funders had no role in study design, data collection and analysis, decision to publish, or preparation of the manuscript.

### Grant Disclosures
The following grant information was disclosed by the authors:
National Key R&D Program of China: 2023YFB2704601.

### Competing Interests
The authors declare that they have no competing interests.

### Author Contributions
- Hafiz Muhammad Qadir conceived and designed the experiments, authored or reviewed drafts of the article, and approved the final draft.
- Muhammad Taseer Suleman conceived and designed the experiments, prepared figures and/or tables, and approved the final draft.
- Rafaqat Alam Khan conceived and designed the experiments, performed the experiments, performed the computation work, prepared figures and/or tables, and approved the final draft.
- Jianqiang Li analyzed the data, authored or reviewed drafts of the article, and approved the final draft.
- Tariq Mahmood performed the experiments, prepared figures and/or tables, and approved the final draft.
- Tanzila Saba analyzed the data, authored or reviewed drafts of the article, and approved the final draft.

## Data Availability

The smog data and code is available at Zenodo:

Rafaqatkhan, K. rafaqatkhan-ai/smog: Initial release of Smog Dataset and Code (Code and Data): https://doi.org/10.5281/zenodo.16790432.

## Supplemental Information

Supplemental information for this article can be found online at http://dx.doi.org/10.7717/peerj-cs.3162#supplemental-information.

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
