# Peer review of "Leveraging deep learning and ensemble learning for air quality forecasting in smart urban environment"

_PeerJ Computer Science, doi:10.7717/peerj-cs.3162_

## Round 0.1 · original submission · Minor Revisions

Please go through the reviews and make necessary changes. Once you incorporate all the changes, please submit the new version for the final accept/reject.

**Language Note:** The review process has identified that the English language must be improved. PeerJ can provide language editing services - please contact us at [email protected] for pricing (be sure to provide your manuscript number and title). Alternatively, you should make your own arrangements to improve the language quality and provide details in your response letter. – PeerJ Staff

·

Basic reporting

This paper is generally well-written. The introduction provides a strong contextual foundation, highlighting the critical issue of urban air pollution, particularly in developing nations.

The literature review is thorough and relevant. It references a wide array of current works from credible journals. The integration of recent studies strengthens the manuscript’s credibility.

The paper adheres well to PeerJ structure and disciplinary standards. Sections are logically organized with smooth transitions between Introduction, Literature Review, Methodology, Results, and Conclusion.

Experimental design

The study uses accuracy as the primary metric, which is suitable given the balanced dataset. The use of confusion matrices for multiclass smog level classification adds interpretability.

Some points should be improved, include a justification for excluding other metrics like F1-score, precision, or recall. Also, no hyperparameter tuning was applied.

Validity of the findings

The study presents a robust experimental framework, testing seven machine learning and deep learning models across four years.

There are some points, The extremely high accuracy of models like Decision Tree and Random Forest raises the possibility of overfitting, yet the paper does not discuss this risk or assess generalization beyond the test set.

Although the reported model accuracies are high, no statistical tests such as t-tests or confidence intervals are provided to confirm whether the differences between models are statistically significant.

Additional comments

The authors effectively combine deep learning and ensemble learning techniques. The experimental setup is comprehensive, covering multiple models and analyzing performance over multiple years, which enhances the practical value of the study.

The data processing pipeline, training methodology, and evaluation procedures are generally clear. The inclusion of code and dataset access points improves the reproducibility and openness of the research.

In its current form, the paper would benefit from revisions. Specifically, the authors should consider incorporating additional performance metrics such as recall and F1-score, address the risk of overfitting with appropriate statistical analysis.

Reviewer 2 ·

Basic reporting

No comment

Experimental design

No comment

Validity of the findings

Lack of Novelty

Reviewer 3 ·

Basic reporting

To start with, the manuscript is written in proper English, it is informative and pleasant to read. The text flows smoothly, so are the following sections and subsections - the paper is properly formatted.
The topic is interesting, practical and potentially lifesaving - many people could benefit from better air quality, which would lead to higher quality of life of individuals.
Personally, I tend to write Interent with a Capital I, just like Bluetooth, as it is the name of a technology.
The number and quality of cited references in the theoretical part is sufficient.
Check whether all abbreviations and acronyms appearing in text are fully described when first mentioned.

Experimental design

Methods are properly described, they are understandable and properly justified.

Validity of the findings

Figures and presented results could be inserted in higher resolution, with larger fonts and sharp lines, making it easier to read and interpret. Do check whether a different file format would be more suitable to present them.
Think especially of a different layout for figs. 6, 8 - the current form is not acceptable.

·

Basic reporting

Strengths:

1. The paper is well-structured and adheres to the general research publishing expectations.
2. Figures and tables support the outcome and the results highlighted.
3. This research addresses a highly relevant area urban air pollution.

Areas for Improvement:

The manuscript contains multiple grammatical errors. An extensive proof-read and an edit by a native English speaker is recommended.
Ex:
1. “This escalating pollution has become a global issue posing…” (Line 34) – needs smoother phrasing

Literature Review:

1. References highlighted in the paper are current and relevant but they lack critical comparison with the current day study’s contributions.

Figures and Tables:

1. Spiral visualizations (Figures 5 and 7) are helpful and relevant, but they lack explanation in the main text.

Experimental design

Strengths:

1. The paper highlights a comparison of traditional machine learning and deep learning models (SVM, KNN, CNN, DNN, LSTM, Random Forest, Decision Tree).
2. Data volume (~26,000 records over 4 years) is robust, and the 80:20 split is appropriate.

Concerns:

Reproducibility: The methods highlighted would benefit from more detailed explanation of model architecture (e.g., CNN and LSTM layers, dropout rates, epochs, batch sizes).

Limitations: The model does not account for critical variables (e.g., temperature, humidity, wind speed) which has an impact on predictive performance. This is highlighted only in the conclusion and should be better integrated into the manuscript.

Validity of the findings

Strengths:

1. Results highlight the accuracy for both PM2.5 and PM10, particularly using Random Forest and Decision Tree models.
2. The multi-year testing across various models is a great validation strategy.

Limitations and Areas to Improve:
1. Deep learning models like CNN and LSTM are discussed, however, their advantage over simpler models is not demonstrated.
2. This paper mentions geographic and feature limitations in the conclusion. These would be better addressed earlier (possibly in the Methodology or Discussion sections) with suggestions.

Additional comments

1. Correct the language and grammar throughout.
2. Elaborate the discussion on limitations and generalization, especially since the dataset is localized (Lahore only).
3. Include more comprehensive metrics to support performance.

---

## Round 0.2 · accepted · Accept

Thanks for incorporating all the suggested changes. This version is a significant improvement.